# Effect of *N*-Acetylcysteine Administration on 30-Day Mortality in Critically Ill Patients with Septic Shock Caused by Carbapenem-Resistant *Klebsiella pneumoniae* and *Acinetobacter baumannii*: A Retrospective Case-Control Study

**DOI:** 10.3390/antibiotics10030271

**Published:** 2021-03-08

**Authors:** Alessandra Oliva, Alessandro Bianchi, Alessandro Russo, Giancarlo Ceccarelli, Francesca Cancelli, Fulvio Aloj, Danilo Alunni Fegatelli, Claudio Maria Mastroianni, Mario Venditti

**Affiliations:** 1Department of Public Health and Infectious Diseases, Sapienza University of Rome, 00185 Rome, Italy; alessandro.bianchi@uniroma1.it (A.B.); alessandro.russo1982@gmail.com (A.R.); giancarlo.ceccarelli@uniroma1.it (G.C.); francesca.cancelli@uniroma1.it (F.C.); claudio.mastroianni@uniroma1.it (C.M.M.); mario.venditti@uniroma1.it (M.V.); 2IRCCS Neuromed, Istituto Neurologico Mediterraneo, 86077 Pozzilli (IS), Italy; aloj@neuromed.it; 3Department of Statistical Science, Sapienza University of Rome, 00185 Rome, Italy; danilo.alunnifegatelli@uniroma1.it

**Keywords:** carbapenem-resistant *Klebsiella pneumoniae*, carbapenem-resistant *Acinetobacter baumannii*, *N*-acetylcysteine, septic shock, critically ill patients

## Abstract

Carbapenem-resistant *Klebsiella pneumoniae* (CR-Kp) and *Acinetobacter baumannii* (CR-Ab) represent important cause of severe infections in intensive care unit (ICU) patients. *N*-Acetylcysteine (NAC) is a mucolytic agent with antioxidant and anti-inflammatory properties, showing also in-vitro antibacterial activity. Aim was to evaluate the effect on 30-day mortality of the addition of intravenous NAC to antibiotics in ICU patients with CR-Kp or CR-Ab septic shock. A retrospective, observational case:control study (1:2) in patients with septic shock caused by CR-Kp or CR-Ab hospitalized in two different ICUs was conducted. Cases included patients receiving NAC plus antimicrobials, controls included patients not receiving NAC. Cases and controls were matched for age, SAPS II, causative agent and source of infection. No differences in age, sex, SAPS II score or time to initiate definitive therapy were observed between cases and controls. Pneumonia and bacteremia were the leading infections. Overall, mortality was 48.9% (33.3% vs. 56.7% in cases and controls, *p* = 0.05). Independent risk factors for mortality were not receiving NAC (*p* = 0.002) and CR-Ab (*p* = 0.034) whereas therapy with two in-vitro active antibiotics (*p* = 0.014) and time to initial definite therapy (*p* = 0.026) were protective. NAC plus antibiotics might reduce the 30-day mortality rate in ICU patients with CR-Kp and CR-Ab septic shock.

## 1. Introduction

Carbapenem-resistant *Klebsiella pneumoniae* (CR-Kp) and *Acinetobacter baumannii* (CR-Ab) represent nowadays an important cause of severe infections in intensive care unit (ICU) patients and mortality rates are significantly associated to septic shock [1,2,3,4,5,6]. Protective factors that influence the clinical outcome include early appropriate antibiotic treatment, adequate source control and number of in-vitro active antimicrobials, whereas septic shock caused by CR-Ab might exhibit a mortality rate up to 60% [6,7]. Therefore, in the context of increasing antimicrobial resistance and restricted therapeutic options typical of the contemporary era, there is a growing scientific interest on finding possible therapeutic adjuvants for sepsis and septic shock [8,9,10,11]. Since septic shock is characterized by excessive and unbalanced production of pro-inflammatory cytokines, reactive oxygen species and a marked alteration of circulation, compounds able to counteract these effects might find a rationale in the treatment of this condition [12,13,14,15,16,17].

*N*-Acetylcysteine (NAC) is a mucolytic agent with antioxidant and anti-inflammatory properties, commonly used for the treatment of acetaminophen overdose or respiratory conditions with high mucus production [18,19,20]. Beyond this, NAC showed also in-vitro activity against several bacteria including multi-drug resistant (MDR) ones and viruses and demonstrated a synergistic interaction with antibiotics or antivirals [21,22,23,24,25,26,27,28,29,30]. In addition, animal models showed improvement of organ damage and a reduction of microvascular dysfunction following NAC administration in endotoxin-induced shock [12,31], rendering this compound attractive for the clinical use as a therapeutic adjuvant in case of infections.

To date, clinical studies evaluating NAC in septic shock gave conflicting results; however, most of them were not recent [32,33,34,35,36,37]. On the other hand, a recent randomized clinical trial investigating the effect of different anti-oxidants as adjuvants in septic shock showed that NAC was able to improve antioxidant capacity [38].

Besides the common use of NAC in the clinical practice, currently in some Italian centers including the ICU of IRCCS Neuromed (Pozzilli, Italy), intravenous NAC is routinely administered in critically ill patients with respiratory conditions characterized by excessive and/or thick mucus production.

Therefore, based on these premises, the purpose of the study was to evaluate the effect on 30-day mortality of the addition of intravenous NAC to antibiotic therapy in ICU patients with septic shock caused by CR-Kp or CR-Ab.

## 2. Results

During the study period, there were 41 cases of patients who had septic shock caused by CR-Kp or CR-Ab treated with NAC. Eleven out of 41 (26.8%) were excluded from the study: central nervous system infections (four cases), no sufficient data (four cases) or no matched controls (three cases). Eventually, 90 patients were enrolled in the study (30 cases and 60 matched controls) (Figure 1).

Mean age was 58.1 and 59.2 years in case and control groups, respectively. 80% of cases and 68.3% of controls were male. SAPS II was 35.3 and 38.6 in cases and controls, respectively. Previous antibiotic therapy was recorded in 63.3% and 48.3% of cases and controls, respectively, whereas a previous CR-Kp or CR-Ab colonization was found in 43.3% and 28.3% of cases and controls, respectively. Length of ICU stay was statistically significant longer in cases than in controls (51.4 vs. 27.8 days, *p* < 0.001). Study population characteristics are shown in Table 1.

In both groups pneumonia was the most frequent source of infection (66.7%), followed by primary bacteremia (33.3%). As for causative agent, 60% and 40% of patients had a septic shock caused by CR-Kp and CR-Ab, respectively. Colistin resistant strains represented 26.7% and 31.7% of isolates in case and control group, respectively, without statistical differences.

Combination therapy was used in almost all patients. In case group, 40% (12/30) of patients received a combination of two antibiotics, 43.3% (13/30) a combination of three antibiotics and 13.3% (4/30) a combination of four antibiotics. A definitive antibiotic regimen containing colistin and/or carbapenem was the most commonly used, respectively in 80% and 73.3% of cases, followed by regimens containing tigecycline (23.3%) and aminoglycoside (20%). In control group, 8.3% (5/60) of patients received monotherapy, 31.6% (19/60) of patients received a combination of two antibiotics, 46.7% (28/60) a combination of three antibiotics and 11.7% (7/60) a combination of four antibiotics. A definitive antibiotic regimen containing carbapenem (73.3%) was the most used, followed by regimens containing colistin (55%), tigecycline (55%) and aminoglycoside (13.3%). Rifampin-containing regimens were used in 6.7% and 20% in cases and controls, respectively (*p* = 0.12). No differences were observed in the two study groups regarding the use of carbapenems, while colistin-containing regimen was used more frequently in cases than in controls (*p* = 0.02). Conversely, tigecycline-containing regimen was used more frequently in controls (*p* = 0.007) (Table 1).

Time to initiate definitive antibiotic therapy was 2.7 days for both groups. Length of antibiotic therapy was similar in the two groups, 15.1 days for cases and 12.3 days for controls (*p* = 0.12).

In the first 24 h from septic shock onset, treatment with two or more antibiotics displaying in vitro activity was reported in 16.7% of cases and in 23.3% of controls and definitive therapy with two or more antibiotics displaying in vitro activity was reported in 20% of cases and 26.7% of controls, without statistical differences. The mean (± SD) administered NAC dosage was 1520 ± 504 mg/die, ranging from 1200 to 3000 mg/die, according to treating physicians. Mean (± SD) duration of NAC treatment was 16.6 ± 7.1 days and no adverse events were recorded during NAC administration.

Overall 7-day (13.3% in cases, 25% in controls) and 14-day (20% in cases, 31.7% in controls) mortality rates were lower in cases than controls, without reaching the statistical significance. On the other hand, the 30-day mortality rate (48.9%) was lower in cases than controls at univariate analysis (33.3% in cases versus 56.7% in controls, *p* = 0.05). Figure 2 shows the 30-day overall survival rate in cases and controls.

In addition, mortality was higher when septic shock was caused by CR-Ab [22/36 (61.1%) versus 22/54 (40.7%) in CR-Ab and CR-Kp, respectively].

At the univariate analysis, risk factor for mortality were age (*p* = 0.01), CR-Ab infection (*p* < 0.001), not receiving NAC (*p* = 0.05), whereas time to initiate definitive therapy (*p* = 0.017) and definitive therapy with two or more antibiotics displaying in vitro activity (*p* = 0.005) were protective.

At the multivariate analysis, independent risk factors for mortality were not receiving NAC (HR: 3.6; 95% CI, 1.59 to 8.22; *p* = 0.002) and CR-Ab infection (HR: 2.8; 95% CI, 1.08 to 7.24; *p* = 0.034); whereas time to initiate definitive therapy (HR: 0.83; 95% CI, 0.70 to 0.98; *p* = 0.026) and definitive therapy with two or more antibiotics displaying in vitro activity (HR: 0.21; 95% CI, 0.06 to 0.73; *p* = 0.014) were protective, regardless of age, sex, SAPS II score, source of infection or the type of antibiotics used as definitive therapy (Table 2).

## 3. Discussion

Septic shock is associated with high mortality rate, particularly when caused by CR Kp or CR Ab [5,6,39,40] being the latter associated to a worse prognosis [39]. To the best of our knowledge, this case-control study analyzed for the first time the effects on 30-day mortality of NAC administration in addition to antibiotic therapy in critically ill patients with septic shock due to CR-Kp or CR-Ab. With this regard, we were able to demonstrate that in patients who received NAC, 30-day mortality was significantly lower than in controls.

NAC is the N-acetyl derivative of the amino acid L-cysteine with anti-oxidant properties thanks to the increase of glutathione in the body, able to reduce free oxygen radicals and to inhibit the effect of pro-inflammatory cytokines [41]. Additionally, NAC has also a vasodilatation activity on microcirculation that improves locoregional blood flow [17]. All of the abovementioned phenomena may have important implications in the setting of a dysregulated host response to infection with a high release of pro-inflammatory cytokines, reactive oxygen species and a profound alteration of microcirculation, as it occurs in septic shock [13,14,15,16]. In animal models, it was demonstrated that NAC ameliorates endotoxin shock-induced organ damage through the reduction of free radicals and inflammatory cytokines production [31]. Of note, this effect was observed when NAC was administered either as a pre-treatment or as a post-treatment drug [31] Furthermore, several studies demonstrated in vitro activity of NAC against a large variety of microorganisms, including *K. pneumoniae* and *A. baumannii* [21,22,23,26]. In particular, preliminary results from our group showed that NAC was highly synergic with meropenem against clinical strains of CR-Kp and CR-Ab whereas Pollini et al. found a remarkable synergism of colistin/NAC combinations against CR-Ab [21,22,26].

However, in spite of the promising results from both in vitro and animal studies, human studies might suggest otherwise. On one side, some NAC studies in patients with sepsis showed encouraging results as far as improved tissue oxygenation and hepatosplanchnic flow, decreased oxidative damage and reduction in IL-8 blood concentrations are concerned [32,33,34,35,36]. On the other hand, a meta-analysis of 41 randomized clinical trials investigating the role of NAC on clinical outcomes in sepsis patients showed no benefit on mortality, length of stay, duration of mechanical ventilation, and incidence of new organ failure with early or late NAC administration [37]. Rather, the latter was associated with hemodynamic instability. As a consequence, the authors concluded that clinicians should not routinely use intravenous NAC in sepsis. The conclusion was confirmed even after subgroups analysis between studies focusing on systemic inflammatory response syndrome or sepsis/septic shock. However, when looking in depth within the meta-analysis, all studies referred to the period 1991–2009 and heterogeneity among study populations was also observed [37]. Furthermore, in almost all studies NAC was given after the sepsis syndrome had been established, potentially too late to be beneficial on outcome. Additionally, another potentially important concern was that high doses have been used in the majority of the abovementioned randomized clinical trials, namely 150 mg/kg, which may allow formation of toxic intermediate molecules interfering with potential benefits of NAC therapy [42].

Conversely, in our study NAC was administered at the very early phase of infection before the development of sepsis syndrome and the mean dosage of NAC (1520 mg/die) was lower than that used in previous study, thus reducing the risk of toxic intermediates production.

Finally, a recent randomized clinical trial investigating the effect of different anti-oxidants in septic shock showed that NAC was able to improve antioxidant capacity, in the absence of significant adverse reactions or side effects [38].

Early antibiotic therapy represents a cornerstone of critical care management in septic shock patients [4,40,43,44,45,46]. Accordingly, in our study a protective effect on mortality was related to the time to initiate definitive therapy, defined as the time between infection onset and initial definitive in vitro active therapy. Combination therapy was the most used treatment in our study; however, previous studies suggested that the key factor for decreasing the mortality is not the number of drugs used but rather the administration of at least two in vitro-active antibiotics, in particular for CR-Kp [6,46,47]. Treatment with two or more in vitro-active antibiotics is difficult to achieve in the presence of MDR bacteria because limited options are available to treat these infections, especially for CR-Ab. In fact, in our study only 36.7% of patients were treated with two or more antibiotics showing in vitro activity against the isolates and this probably explains the high mortality rates observed in our population, which is, however, similar to that reported in the literature [6]. Nevertheless, treatment with two or more in vitro-active antibiotics was associated with lower mortality.

Although a difference was observed in the two study groups regarding the use of regimens containing colistin and tigecycline, with the former more frequently used in cases than controls, this did not affect overall mortality.

In our study, 30-day mortality for CR-Ab was higher than that observed for CR-Kp. The impact of CR-Ab on clinical outcome was also highlighted in multivariable analysis, thus confirming the recent literature data, which showed a mortality rate of up to 60% and 40% in the presence of septic shock caused by CR-Ab and CR-Kp, respectively [5,6,48].

Our study has some limitations that should be acknowledged. First, the retrospective nature of the study is an intrinsic limitation of this analysis. Second, the sample size is relatively low and included all patients with septic shock caused by CR-Kp or CR-Ab observed in the two ICUs, without a sample size calculation. Therefore, the results of our study might be considered preliminary and further multicenter prospective studies are needed to confirm our findings. Third, cases and controls came from two distinct cohorts of patients and therefore differences could be related to intrinsic differences between populations and microorganism, despite the extensive patients’ matching (age, SAPS II score, causative agent and source of infection). However, we were confident that the potential differences were minimized by the fact that both ICUs had a dedicated Infectious Diseases consultant who was in charge for the treatment of all patients and who belonged to the same well-established consultation system. Consequently, the Infectious Diseases consultants had the same diagnostic and therapeutic approach towards patients admitted to these 2 ICUs (Appendix A). Fourth, NAC was not administered in all cases at the same dosage and timing and consequently the results need to be interpreted cautiously. Lastly, since study population included mainly patients who had pneumonia, conclusions should mostly apply to ICU patients with pneumonia-associated septic shock caused by CR-Kp and CR-Ab. An additional limitation was the lack of oxidative markers (i.e nitrate/nitrite ratio, glutathione) measurements before and after NAC therapy.

Nevertheless, we believe that the present investigation has some important strenghts, which might have contributed to bias reduction, such as: (i) patients were matched for several variables, which might have had theirselves an influence on the primary outcome of the study (30-day mortality); (ii) each ICU had a dedicated infectious diseases consultant referring to the same well-established consultation system, thus assuring the same clinical and therapeutic approach to infections in both cases and controls. Furthermore, it represents a real-life clinical experience providing useful suggestions to clinicians about the management of a difficult-to-treat infection such as septic shock caused by CR-Kp and CR-Ab.

## 4. Materials and Methods

We performed a retrospective, observational case:control study (1:2) in patients with septic shock caused by CR-Kp or CR-Ab hospitalized in two different ICUs [IRCCS Neuromed for cases and Sapienza University (Rome, Italy) for controls, the latter derived from a historical cohort of patients [6]. Both ICUs have a dedicated Infectious Diseases consultant referring to a well-established consultation system at Policlinico Umberto I, Sapienza University of Rome, with the same clinical and therapeutic approach [49,50,51] (Appendix A). Cases included patients with septic shock receiving intravenous NAC in combination with antimicrobials, controls included patients with septic shock not receiving NAC. For every case, two matched controls were randomly selected from patients who did not receive NAC. Cases and controls were matched for age, SAPS II score, causative agent and source of infection. Data collection for cases was blinded for the outcome. Inclusion criteria were: (i) ICU admission, (ii) presence of septic shock during ICU stay, (iii) laboratory documented and confirmed infection by CR-Kp or CR-Ab and iv) intravenous administration of NAC for cases, whereas exclusion criteria were (i) a documented infection localized to the central nervous system at admission or during hospital stay, (ii) age under 18 years old or (iii) missing key data.

The Ethical Commitees approved the study (no. 4547–2017 for Sapienza; approval 20 March 2019 for IRCCS Neuromed) whereas informed consent was waived due to the retrospective nature of the research.

### 4.1. Baseline Assessment

Patient data were extracted from medical records and from hospital computerized databases or clinical charts. The following information was reviewed: demographics, clinical and laboratory findings, comorbid conditions, microbiological data, duration of ICU and hospital stay, incidence of infections during hospitalization, treatments and procedures administered during hospitalization and/or in the 90 days prior to infection, classes of antibiotics received on admission and/or after admission before a positive culture of a biological sample was obtained, the simplified acute physiology score II (SAPS II) at the time of infection, source of infection, antibiotic regimens used for CR-Ab or CR-Kp infections, and 30-day mortality. According to both hospital’s guidelines, colonization with CR-Kp and CR-Ab strains was routinely evaluated by rectal swab, respiratory specimens and urine culture at the time of ICU admission and every week afterwards.

### 4.2. Definitions

Infections were defined according to the standard definitions of the ECDC [52] and septic shock was defined according to the SEPSIS-3 criteria definition, a subset of sepsis with persisting hypotension requiring vasopressors to maintain mean arterial pressure of 65 mmHg or greater and having a serum lactate level greater than 2 mmol/L despite adequate volume resuscitation [53].

A CR-Ab or CR-Kp infection was defined as clinical signs of infection and culture of blood, urine, cerebrospinal fluid or a biological sample from skin and skin structures, lung, or abdomen yielding a CR-Ab or a CR-Kp strain.

Infection onset was defined as the date of collection of the index culture (i.e., the first blood culture that yielded the study isolate). Infections were classified as hospital acquired if the index culture had been collected > 48 h after hospital admission and no signs or symptoms of infection had been noted at admission. Primary bloodstream infection (BSI) was defined as BSI occurring in patients without a recognized source of infection.

The severity of clinical conditions was determined by using SAPS II score calculated at the time of septic shock onset. Length of hospital and ICU stay were calculated as the number of days from the date of admission to the date of discharge or death.

Depending on the number of drugs used (one or more), treatment regimens were classified as monotherapy or combination therapy. Definitive antibiotic therapy was defined as the definitive antimicrobial treatment based on in vitro CR-Ab or CR-Kp isolates susceptibilities. Antibiotic regimens were also classified according to the following: one antibiotic displaying in vitro activity, and two or more antibiotics displaying in vitro activity. Time to initial definitive therapy was the time between infection onset and initial definitive therapy.

Intravenous NAC was administered in patients with respiratory conditions characterized by excessive and/or thick mucus production as soon as signs of a possible infection developed (i.e., at the very early phase of infection) as adjunctive therapy and stopped together with antibiotic therapy. Dosages of intravenous NAC ranged from 1200 to 3000 mg/die, according to treating physicians. Intravenous NAC was administered in saline solution with 30–60 min infusion rate.

### 4.3. Statistical Analysis

Continuous variables were compared using Student’s t test or Mann-Whitney U test and were described as mean ± standard deviation (SD) or as median and interquartile range (IQR) according to whether the distribution of the variables was normal or non-normal. Chi-squared test (χ2) and Fisher’s exact test were used to compare categorical variables. Univariate and multivariate analyses were performed to evaluate factors related to 30-days mortality. Variables with a *p* value two-sided <0.05 were considered statistically significant. The results obtained were analyzed using a commercially available statistical software package (version 15, STATA Corp, College Station, TX, USA: StataCorp LLC).

## 5. Conclusions

In conclusion, in the challenging context of increasing antimicrobial resistance and restricted therapeutic options, this study suggests that a combined use of NAC plus antibiotics might reduce the 30-day mortality rate in ICU patients with septic shock caused by CR-Kp and CR-Ab. Therefore, our preliminary data seem to encourage further clinical investigations on the role of NAC as an adjuvant therapy in ICU patients with septic shock due to multi-drug resistant Gram-negative bacilli.

## Figures and Tables

**Figure 1 antibiotics-10-00271-f001:**
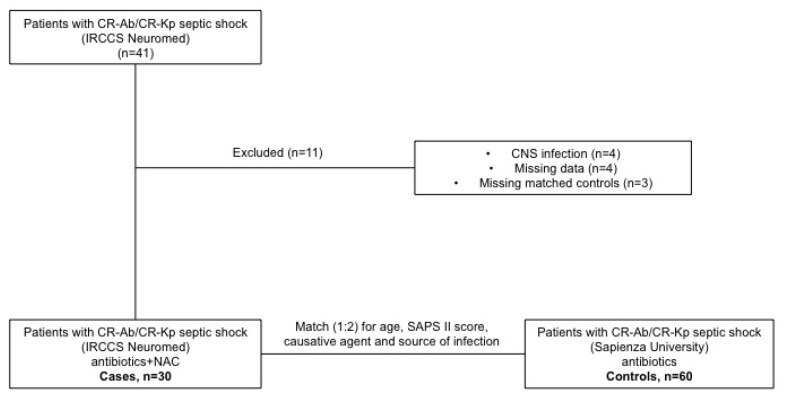
Flow-chart of the study.

**Figure 2 antibiotics-10-00271-f002:**
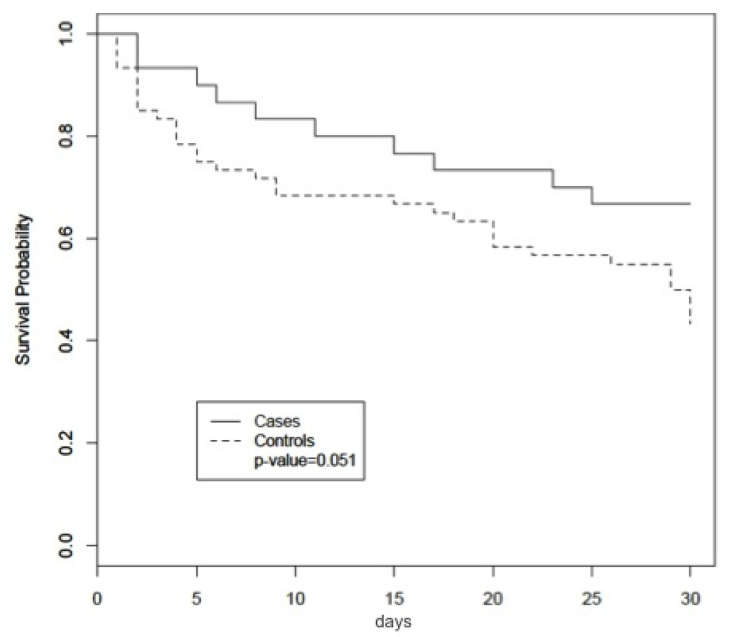
Overall 30-day survival rate in patients with carbapenem-resistant *K. pneumoniae* or *A. baumannii* septic shock receiving antibiotics plus intravenous NAC (cases, n = 30) or antibiotics only (controls, n = 60).

**Table 1 antibiotics-10-00271-t001:** Characteristics of patients with septic shock caused by carbapenem-resistant *Klebsiella pneumoniae* (CR-Kp) and *Acinetobacter baumannii* (CR-Ab).

	Cases ° n = 30	Controls ° n = 60	*p*-Value
**Age, years (mean ± SD)**	58.1 ± 17.7	59.2 ± 14.19	*
**Male sex, n (%)**	24 (80)	41 (68.3)	0.32
**SAPS II**	35.33 ± 17.7	38.57 ± 11.5	*
**Lenght of ICU stay, days (mean ± SD)**	51.4 ± 27.9	27.8 ± 20	<0.0001
**Previous (90-d) hospitalization, n (%)**	14 (46.6)	21 (35)	0.36
**Previous (90-d) ICU admission, n (%)**	4 (13.3)	6 (10)	0.72
**Previous (90-d) surgery, n (%)**	14 (43.3)	17 (28.3)	0.16
**Previous (90-d) antibiotic therapy, n (%)**	19 (63.3)	29 (48.3)	0.26
**Previous colonization with CR-Kp or CR-Ab, n (%)**	13 (43.3)	17 (28.3)	0.16
**Comorbidities, n (%)** - chronic liver disease - neoplasm - diabetes mellitus - cardiovascular diseases - chronic renal failure - COPD	6 (20) 6 (20) 6 (20) 18 (60) 0 (0) 3 (10)	7 (11.6) 2 (3.3) 15 (25) 19 (31.6) 2 (3.3) 11 (18.3)	0.34 0.01 0.79 0.01 0.55 0.37
**Causes of ICU admission, n (%)** - respiratory failure - septic shock - stroke - post-surgery - trauma - cardiac arrest	6 (20) 3 (10) 12 (40) 6 (20) 2 (6.6) 1 (3.3)	20 (33.3) 14 (23.3) 4 (6.6) 7 (11.6) 9 (15) 6 (10)	0.22 0.16 0.0002 0.34 0.32 0.41
**Source of infection, n (%)** - pneumonia - primary bacteremia	20 (66.7) 10 (33.3)	40 (66.7) 20 (33.3)	*
**Causative agent, n (%)** - CR-Kp - CR-Ab	18 (60) 12 (40)	36 (60) 24 (40)	*
**Colistin-resistant strains, n (%)**	8 (26.7)	29 (31.7)	0.54
**Adequate source control, n (%)**	10 (33.3)	31 (51.6)	0.12
**Number of antibiotics used as definitive therapy, n (%)** - no definite therapy - 1 antibiotic - 2 antibiotics - 3 antibiotics - 4 antibiotics	1 (3.4) 0 (0) 12 (40) 13 (43.3) 4 (13.3)	1 (1.7) 5 (8.3) 19 (31.6) 28 (46.7) 7 (11.7)	0.99 0.16 0.48 0.82 0.99
**Type of antimicrobial combinations, n (%)** - Carbapenem-containing regimen - Colistin-containing regimen - Tigecycline-containing regimen - Aminoglycoside-containing regimen - Rifampin-containing regimen	22 (73.3) 24 (80) 7 (23.3) 6 (20) 2 (6.7)	44 (73.3) 33 (55) 33 (55) 8 (13.33) 13 (20)	0.99 0.02 0.007 0.53 0.12
**≥2 in-vitro active antibiotics within 24 h** **from septic shock, n (%)**	5 (16.7)	14 (23.3)	0.58
**≥2 in-vitro active antibiotics definitive, n (%)**	6 (20)	16 (26.7)	0.60
**Time to initial definitive therapy, days (mean ± SD)**	2.7 ± 0.4	2.65 ± 0.2	0.86
**NAC dosage, mg/die (mean** **± SD)** **Range**	1520 ± 504 (1200–3000)	NA	
**Length of antibiotic therapy, days (mean** **± SD)**	15.1 ± 7.9	12.3 ± 8.3	0.12
**Length of NAC therapy, days (mean** **± SD)**	16.6 ± 7.1	NA	
**Adverse effects of NAC therapy, n(%)**	0 (0)	NA	
**Outcome, n (%)** - 7-day mortality - 14-day mortality - 30-day mortality	4 (13.3) 6 (20) 10 (33.3)	15 (25) 19 (31.7) 34 (56.7)	0.18 0.32 0.051

°: Cases included patients receiving intravenous NAC in combination with antimicrobials, controls included patients not receiving NAC. Data collection for cases was blinded for the outcome. *: Cases and controls were matched for age, SAPS II, source of infection and causative agent. ICU: Intensive Care Unit. CR-Kp: Carbapenem-resistant *Klebsiella pneumoniae*; CR-Ab: Carbapenem-resistant *Acinetobacter baumannii*; COPD: Chronic Obstructive Pulmonary Disease. NA: not applicable.

**Table 2 antibiotics-10-00271-t002:** Indipendent risk factors for 30-day mortality of patients with septic shock caused by carbapenem-resistant *Klebsiella pneumoniae* and *Acinetobacter baumannii*.

Variable	Univariate Analysis			Multivariate Analysis		
	HR	95% CI	*p*-Value	HR	95% CI	*p*-Value
Controls (not receiving NAC)	1.99	0.98–4.04	0.05	3.61	1.59–8.22	0.002
Sex	0.88	0.46–1.68	0.70	1.35	0.68–2.69	0.38
Age	1.02	1.00–1.05	0.015	1.02	0.99–1.05	0.17
SAPS II	1.01	0.99–1.03	0.30	1.01	0.98–1.05	0.25
CR-Ab	3.29	1.81–6.00	<0.001	2.79	1.07–7.24	0.03
≥2 in-vitro active antibiotics	0.22	0.08–0.63	0.005	0.21	0.06–0.73	0.014
Number of antibiotics in definitive therapy	0.69	0.47–1.03	0.07	0.65	0.39–1.09	0.10
Pneumonia	1.74	0.86–3.53	0.12	0.79	0.32–1.94	0.62
Use of colistin	0.94	0.51–1.73	0.86	0.50	0.19–1.32	0.16
Time to definitive antibiotic therapy	0.81	0.69–0.96	0.01	0.82	0.69–0.97	0.026

NAC: *N*-acetylcysteine; CR-Ab: Carbapenem-resistant *Acinetobacter baumannii.* SAPS II: Simplified Acute Physiology Score II. CI: Confidence interval.

## Data Availability

The data used to support the findings of this study are available from the corresponding author upon request.

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
