# Peer review of "Effect of N-Acetylcysteine Administration on 30-Day Mortality in Critically Ill Patients with Septic Shock Caused by Carbapenem-Resistant Klebsiella pneumoniae and Acinetobacter baumannii: A Retrospective Case-Control Study"

_antibiotics, 2021, doi:10.3390/antibiotics10030271_

Round 1

Reviewer 1 Report

Thank you for the opportunity to review the study on the “Effect of N-acetylcysteine administration on 30-day mortality in critically ill patients with septic shock caused by carbapenem-resistant Klebsiella pneumoniae and Acinetobacter baumannii: a case-control study”. The study's purpose of the study was to evaluate the effect on 30-day mortality of the addition of intravenous NAC to antibiotic therapy in 62 ICU patients with septic shock caused by CR-Kp or CR-Ab. Overall, the article is well-written and the finding is interesting. However, a few issues need to be addressed in their current form.

These are the comments for the authors:

A randomized controlled study is a preferable method to study drug efficacy, however, a case-controlled study in some situations is acceptable.

The study selected controls from other ICU, could the author justify the reason? Are both ICU using a similar treatment regime? The treatment regimes are one of the major confounder factors that affect 30-days mortality. Hence, The authors shall spell out the treatment protocol as supplementary material.

There are significant differences in the antimicrobial regimes (ie. Colistin-containing & Tigecycline-containing regime) between the cases and controls group. Will these significant differences possibly affect the mortality outcome? The author should discuss this interesting further.

The study may seem underpowered with a low sample size and insufficient matching ratio. Due to the nature of the study is about treatment recommendation, the authors should justify this by providing detailed data on sample size calculation to convince the reader. This is of utmost importance for this paper.

Result

The results are well presented and written.

However, due to the heterogeneity of the NAC dosage and timing of delivery, the results need to be interpreted cautiously.

Conclusion

The study population is mainly patients who had pneumonia. The conclusion for this study shall only apply to ICU patients with pneumonia associated septic shock caused by CR-Kp and CR-Ab.

Some minor comments

Typo in line 66: “43”. It should be “41”

Typo in line 133: “ indipendent”.

Author Response

Reviewer1

Thank you for the opportunity to review the study on the “Effect of N-acetylcysteine administration on 30-day mortality in critically ill patients with septic shock caused by carbapenem-resistant Klebsiella pneumoniae and Acinetobacter baumannii: a case-control study”. The study's purpose of the study was to evaluate the effect on 30-day mortality of the addition of intravenous NAC to antibiotic therapy in 62 ICU patients with septic shock caused by CR-Kp or CR-Ab. Overall, the article is well-written and the finding is interesting. However, a few issues need to be addressed in their current form.

 These are the comments for the authors:

A randomized controlled study is a preferable method to study drug efficacy, however, a case-controlled study in some situations is acceptable.

A: Thank you for your comment. Unfortunately, we could not perform a randomized clinical trial, therefore, we considered that a well-matched case:control study could have been acceptable for the purpose of the study.

The study selected controls from other ICU, could the author justify the reason? Are both ICU using a similar treatment regimen? The treatment regimes are one of the major confounder factors that affect 30-days mortality. Hence, The authors shall spell out the treatment protocol as supplementary material.

A: Yes, cases and controls were from 2 different ICUs. In particular one ICU (IRCCS Neuromed) was chosen since intravenous NAC is routinely administered by intensivists in critically ill patients as adjunctive therapy for respiratory and other infections, whereas in  the other ICU (Sapienza University of Rome) NAC was never used. In both the ICUs the dedicated Infectious Diseases consultant belonged to the same well-established consultation system, with the same clinical and therapeutic approach (see also Supplementary Table1). Therefore, the only difference between the two selected ICUs was NAC administration.

We agree with the reviewer that treatment regimes are one of the major confounder factors that affect 30-days mortality. However, we were confident on choosing different ICUs for the study for the following reasons: i) both ICUs had a dedicated Infectious Diseases consultant who was in charge for the treatment of all patients and who belonged to the same well-established consultation system at Policlinico Umberto I, Sapienza University of Rome; ii) belonging to the same well-established consultation system, the Infectious Diseases consultants had the same diagnostic and therapeutic approach towards patients admitted to these 2 ICUs.

Accordingly, we modififed the text as follows: “However, we were confident that the potential differences on treatments were minimized by the fact that both ICUs had a dedicated Infectious Diseases consultant who was in charge for the treatment of all patients and who belonged to the same well-established consultation system. Consequently, the Infectious Diseases consultants had the same diagnostic and therapeutic approach towards patients admitted to these 2 ICUs (Supplementary Figure1).”

We inserted Supplementary Figure1 and added 3 additional references (Ref 49-51).

There are significant differences in the antimicrobial regimens (ie. Colistin-containing & Tigecycline-containing regime) between the cases and controls group. Will these significant differences possibly affect the mortality outcome? The author should discuss this interesting further.

A: Thank you for the comment. Although a difference was observed in the two study groups regarding the use of regimens containing colistin and tigecycline, with the former more frequently used in cases than controls, this did not affect overall mortality. We added this sentence in the discussion.

The study may seem underpowered with a low sample size and insufficient matching ratio. Due to the nature of the study is about treatment recommendation, the authors should justify this by providing detailed data on sample size calculation to convince the reader. This is of utmost importance for this paper.

A: Thank you for the comment. The study was a retrospective one, including all patients with septic shock caused by CR-Ab and CR-Kp observed in the IRCCS Neuromed ICU during the study period (cases) and we performed an extensive match for controls in order to minimize potential bias. Therefore, we were able to perform a 1:2 match for the following characteristics: age, SAPS II score, causative agent and source of infection. Furthermore, data collection for cases was blinded for the outcome.

The results of the our study might be considered preliminary and we also included the lack of a sample size calculation as an additional limitation of the study. We modified the text as follows: “Second, the sample size is relatively low and included all patients with septic shock caused by CR-Kp or CR-Ab observed in the two ICUs, without a sample size calculation. Therefore, the results of our study might be considered preliminary and further multicenter prospective studies are needed to confirm our findings.”

We also modified the title by inserting the term “retrospective”.

Result

The results are well presented and written.

However, due to the heterogeneity of the NAC dosage and timing of delivery, the results need to be interpreted cautiously.

  1. Thank you for the comment. We modified the limitation sections by adding the sentence: “due to the heterogeneity of the NAC dosage and timing of delivery, the results need to be interpreted cautiously.”

Conclusion

The study population is mainly patients who had pneumonia. The conclusion for this study shall only apply to ICU patients with pneumonia associated septic shock caused by CR-Kp and CR-Ab.

  1. Thank you for the comment. We added the following sentence in the limitation part: “Lastly, since study population included mainly patients who had pneumonia, conclusions should mostly apply to ICU patients with pneumonia-associated septic shock caused by CR-Kp and CR-Ab”

 Some minor comments

Typo in line 66: “43”. It should be “41”.

A: We modified the number.

Typo in line 133: “ indipendent”.

A: We inserted the word indipendent.

Reviewer 2 Report

The MS presents a short but important study regarding the effect on 30-day mortality of intravenous NAC on antibiotics treatment  in ICU patients with Carbapenem-resistant Klebsiella pneumoniae (CR-Kp) and Acinetobacter baumannii (CR-Ab) septic shock.

The text is well written in a scientific manner. I appreciate the authors for clearly acknowledging the limitations of the study. The hypothesis and rationale for the study are well mentioned. The flow chart is very helpful to understand the scene of the work. The work conducted methodically. 

Although it's a limited study and may be useful more for the clinician-working in the area of sepsis clinical research. 

Author Response

Reviewer 2

The MS presents a short but important study regarding the effect on 30-day mortality of intravenous NAC on antibiotics treatment  in ICU patients with Carbapenem-resistant Klebsiella pneumoniae (CR-Kp) and Acinetobacter baumannii (CR-Ab) septic shock.

The text is well written in a scientific manner. I appreciate the authors for clearly acknowledging the limitations of the study. The hypothesis and rationale for the study are well mentioned. The flow chart is very helpful to understand the scene of the work. The work conducted methodically. 

Although it's a limited study and may be useful more for the clinician-working in the area of sepsis clinical research. 

A: Thanks for the reviewer’s comments.

Reviewer 3 Report

The authors have put together an interesting study to evaluate the effects on mortality of adding intravenous N-acetylcysteine (NAC) to antibiotic therapy in ICU patients with septic shock caused by carbapenem (CR)-Kp or CR-Ab. While this study is important because of the controversial or negligible role played by anti-oxidant therapy on septic shock patients, the lack of specific details in the materials and methods make comparisons with previous studies difficult. Main concerns: 1) The manuscript was easy to follow and the tables although crowded they contained all the required information. 2) The section of materials and methods was in my opinion very incomplete unless this format is acceptable by this journal 3) For example, it states that “intravenous NAC was administered in patients with respiratory conditions characterized by excessive and/or thick mucus production as soon…. as signs of a possible infection developed” Here is where my problems started, 1) what concentration of NAC was administered? 2) for how long?, and most importantly, what were the before and after values of oxidative markers in these patients such as nitrate/nitrite ratio, glutathione, lipid peroxidation, etc. 4) Did the authors identified any side effects in patients taking NAC intravenously?

Author Response

Reviewer 3

The authors have put together an interesting study to evaluate the effects on mortality of adding intravenous N-acetylcysteine (NAC) to antibiotic therapy in ICU patients with septic shock caused by carbapenem (CR)-Kp or CR-Ab. While this study is important because of the controversial or negligible role played by anti-oxidant therapy on septic shock patients, the lack of specific details in the materials and methods make comparisons with previous studies difficult.

Main concerns:

1) The manuscript was easy to follow and the tables although crowded they contained all the required information.

A: Thank you for the comment. According to reviewer’s suggestion, we included in the table 1 additional informations such as duration of NAC treatment and adverse events.

2) The section of materials and methods was in my opinion very incomplete unless this format is acceptable by this journal

A: According to the following reviewer’s suggestions, we modified the method section.

3) For example, it states that “intravenous NAC was administered in patients with respiratory conditions characterized by excessive and/or thick mucus production as soon…. as signs of a possible infection developed”

Here is where my problems started, 1) what concentration of NAC was administered?

A: Unfortunately, the used NAC concentrations was heterogeneous and varied from 1200 to 3000 mg/die, according to treating physicians. We modified method and result sections and inserted the dosage range in the Table1.

Method: Dosages of intravenous NAC ranged from 1200 to 3000 mg/die, according to treating physicians. Intravenous NAC was administered in saline solution with 30-60 min infusion rate. NAC treatment was started as soon as signs of a possible infection developed (i.e. at the very early phase of infection) as adjunctive therapy and was stopped together with antibitiotic therapy.

Result: The mean administered NAC dosage was 1520±504 mg/die (mean±SD), ranging from 1200 to 3000 mg/die, according to treating physicians. Mean (±SD) duration of NAC treatment was 16.6±7.1 days.

2) for how long?,

A: NAC treatment was started as soon as signs of a possible infection developed (i.e. at the very early phase of infection) as adjunctive therapy and was stopped together with antibitiotics. We inserted the following sentence: Mean (±SD) duration of NAC treatment was 16.6±7.1 days.

Furthermore, we inserted the corresponding row in the Table1.

and most importantly, what were the before and after values of oxidative markers in these patients such as nitrate/nitrite ratio, glutathione, lipid peroxidation, etc.

A: This was a retrospective study with the main aim to evaluate the effect of NAC administration on 30-day mortality. Therefore, the aim was clinical and we were not able to measure all the suggested markers.

We agree with the reviewer that measurements of oxidative markers might have given important informations of NAC activity, therefore we added the following sentence in the limitation part: Another limitation was the lack of oxidative markers (i.e nitrate/nitrite ratio, glutathione) measurements before and after NAC therapy.

4) Did the authors identified any side effects in patients taking NAC intravenously?

A: No adverse events were recorded during NAC administration. We inserted the following sentence in the result section: No adverse events were recorded during NAC administration. Furthermore, we inserted the corresponding row in the Table1.

Round 2

Reviewer 1 Report

The manuscript is well revised. The comments are addressed accordingly. I have no further comment. 

Reviewer 3 Report

The authors have done a good job in addressing this reviewer's concerns. Clarifying that this was a retrospective study explains some of the limitations.